# Respiratory diseases in survivors of adult cancer compared with the general population: a systematic review protocol

Kirsty Andresen ![ORCID],[1] Helena Carreira ![ORCID],[1] Jennifer K Quint ![ORCID],[2] Krishnan Bhaskaran[1]

[1]Non-Communicable Diseases Epidemiology, Faculty of Epidemiology and Population Health, London School of Hygiene and Tropical Medicine, London, UK
[2]National Heart and Lung Institute, Imperial College London, London, UK

**Correspondence to**
Kirsty Andresen;
kirsty.andresen1@lshtm.ac.uk

## ABSTRACT

**Introduction** There is concern that survivors of adult cancers may be at increased risk of respiratory infections and of exacerbations of pre-existing respiratory conditions. Considering the high prevalence of respiratory disease in the general population, increased respiratory disease risk in survivors of adult cancers could translate into an important impact on morbidity and mortality. The aim of this systematic review is to summarise and assess the quality of all studies comparing respiratory outcomes between adult cancer survivors and individuals with no history of cancer.

**Methods and analysis** This systematic literature review will be conducted using Medline, EMBASE and Cochrane. We will include cohort or case–control studies that provide a comparative estimate of the risk of a respiratory disease of interest in survivors of adult cancer against a comparator cohort of cancer-free individuals. No geographic, time or language restrictions will be applied. We will assess the risk of bias using the Scottish Intercollegiate Guidelines Network methodology checklists. Results will be summarised by type of respiratory outcome, cancer type and cancer survivorship definition. If sufficient numbers of homogeneous studies are found, summary measures of association will be calculated using random effects meta-analysis models.

**Ethics and dissemination** Ethical approval is not applicable to our study. The results will be used to identify evidence gaps and priorities for future research to understand respiratory morbidity in survivors of adult cancers and identify possible mitigation strategies. Results from this review will be disseminated to clinical audiences and submitted to a peer-reviewed journal when completed.

**Trial registration number** This study has been registered on PROSPERO (registration number: CRD42022311557).

## INTRODUCTION

Currently, 50% of people diagnosed with cancer are expected to survive for 10 or more years.[1] The proportion is likely to increase further considering the rapid and continual improvements in cancer management and treatment. As more patients survive longer, they may experience additional health challenges during their survivorship. There is evidence that cancer survivors are at an increased risk of developing certain adverse

### STRENGTHS AND LIMITATIONS OF THIS STUDY

⇒ The search terms for respiratory outcomes are comprehensive and clinically reviewed by a respiratory clinician.
⇒ Risk of bias will be formally assessed in a structured and standardised way using the Scottish Intercollegiate Guidelines Network methodology checklists.
⇒ No geographic, time or language restrictions will be applied in an attempt to find all eligible studies.
⇒ The search will be conducted in English language within English databases; thus we may miss relevant articles not captured by English language search engines.

outcomes, such as cardiovascular diseases[2] and mental health issues,[3] that may cause morbidity and impaired quality of life. However, the long-term respiratory consequences of cancer and its treatment are less clear.

Respiratory diseases have an immense worldwide burden of morbidity and mortality, accounting for more than 10% of all disability-adjusted life-years, second only to cardiovascular diseases (including stroke).[4] Furthermore, cancer can weaken the immune system which may increase the risk of respiratory infections.[5] Additionally, survivors of cancer often have a higher health burden. A quarter of patients with cancer have pre-existing respiratory comorbidities[6] and there is concern that cancer and its treatment or shared risk factors may be conducive to worse outcomes than in the general population.[7] Considering the high prevalence of respiratory disease in the general population, even a small increase in respiratory disease risk in cancer survivors will translate into an important impact on morbidity and mortality. Thus, identifying possible strategies to mitigate respiratory morbidity in cancer survivors would suppose an important improvement in absolute terms.

We aim to conduct a systematic review to summarise and assess the quality of the current knowledge from studies comparing survivors of adult cancer to individuals with no history of cancer for respiratory outcomes. We aim to evaluate relative risks of a broad spectrum of respiratory outcomes across a wide range of cancer types, in order to build a complete picture of the evidence on respiratory health during cancer survivorship.

## METHODS AND ANALYSIS

This study protocol has been designed according to the Preferred Reporting Items for Systematic Review and Meta-Analysis Protocols (PRISMA-P), registered in the International Prospective Register of Ongoing Systematic Reviews (PROSPERO) and will be reported according to the PRISMA reporting guidelines. PROSPERO registration number: CRD42022311557.

### Objectives

The specific objectives of this study are:

1. Summarise the current evidence on the association between cancer survivorship and incident respiratory disease outcomes. We will consider the relative risk of incident diagnosis, exacerbations (considered as primary care visits or hospitalisations) and mortality outcomes separately for each respiratory disease of interest between survivors of adult cancer and the cancer-free population.
2. Critically appraise the quality of the available evidence and evaluate the main sources of bias.
3. Identify evidence gaps and priorities for future research on respiratory health in cancer survivors.

### Search strategy

The search will include articles published in Cochrane, Medline and EMBASE through to 8 March 2022. No geographic, time or language restrictions will be applied. We will use OVID search interface to simultaneously search both Medline and EMBASE and we will conduct a separate search within Cochrane. The coverage of EMBASE spans from 1974, Medline from 1946 and Cochrane Reviews from 1996. We do not plan to include grey literature as epidemiological studies providing measures of effect with enough information to be assessable are likely to be published in academic journals. Furthermore, we do not plan to routinely contact all study authors. We will search title, abstract and subject headings using our created search expression. The search expression will include keywords and subject headings (both Medical Subject Headings and Emtree) for the target population (survivors of adult cancer) and the target outcomes (respiratory diseases). As an additional step to identify studies that may have been missed by the initial search strategy, we will examine the reference lists of all studies identified in the initial database search that meet the selection criteria. The list of selected respiratory outcomes and associated search terms were confirmed by

---

**Box 1    Search expression for both Medline and EMBASE***

Search expression

⇒ 1) *Cancer Survivors/ or exp cancer survivor
⇒ 2) (cancer survivor* or 'living with and beyond cancer' or 'living with cancer').mp.
⇒ 3) (respiratory and (outcome* or disease* or infection* or condition* or disorder or manifestation*)).mp.
⇒ 4) (lung and (outcome* or disease* or infection* or condition* or disorder or manifestation*)).mp.
⇒ 5) (pulmonary and (outcome* or disease* or infection* or condition* or disorder or manifestation*)).mp.
⇒ 6) exp respiratory tract disease/ or exp lung disease/
⇒ 7) ('upper respiratory infection*' or 'upper respiratory tract infection*' or nasopharyngitis or rhinitis or pharyngitis or sinusitis or tonsilitis or laryngitis or tracheitis or laryngotracheitis).mp.
⇒ 8) respiratory tract infections/ or common cold/ or empyema, pleural/ or laryngitis/ or pharyngitis/ or rhinitis/ or sinusitis/ or whooping cough/ or exp upper respiratory tract/ or exp upper respiratory tract congestion/ or exp upper respiratory tract infection/
⇒ 9) bronchitis.mp. or Bronchitis/ or Infectious bronchitis virus/
⇒ 10) pulmonary diseases/ or chronic obstructive/ or bronchitis, chronic/ or pulmonary emphysema/ or Bronchitis, Chronic/ or exp chronic bronchitis/ or exp bronchitis/
⇒ 11) (influenza* or fluinfluenza).mp. or influenza, human/
⇒ 12) exp influenza/
⇒ 13) Pneumonia/ or pneumonia.mp.
⇒ 14) exp pneumonia/
⇒ 15) Asthma/ or asthma.mp.
⇒ 16) exp asthma/
⇒ 17) (chronic obstructive pulmonary disease or COPD or chronic bronchitis or emphysema).mp.
⇒ 18) exp chronic obstructive lung disease/
⇒ 19) interstitial lungs disease.mp. or Lung Diseases, Interstitial/
⇒ 20) exp interstitial lung disease/
⇒ 21) Pulmonary Fibrosis/
⇒ 22) (pulmonary fibrosis or lung fibrosis).mp.
⇒ 23) exp lung fibrosis/
⇒ 24) pneumonitis.mp.
⇒ 25) exp coronavirus disease 2019/or exp COVID-19/or (COVID-19 or SARS-CoV-2 or coronavirus 19).mp.
⇒ 26) 3 or 4 or 5 or 6 or 7 or 8 or 9 or 10 or 11 or 12 or 13 or 14 or 15 or 16 or 17 or 18 or 19 or 20 or 21 or 22 or 23 or 24 or 25
⇒ 27) 1 or 2
⇒ 28) 26 and 27

*Both MeSH terms and Emtree terms were included in our search expression.

---

a respiratory consultant (JKQ). The search expression is included in box 1.

We defined respiratory disease as any upper or lower respiratory condition. See table 1 for list of included respiratory diseases.

### Screening

The screening will be conducted in two stages:

1. Screening of titles and abstracts: titles and abstracts will be screened against the selection criteria by one reviewer. A second reviewer will screen a random sample of 10% of articles identified in the initial search. Any discrepancies will be resolved by discussion between

**Table 1** Categorisation of respiratory diseases included in the systematic literature review

| Category | Conditions of interest |
|---|---|
| COVID-19 | Infection with SARS-CoV-19 |
| Upper respiratory infections | Nasopharyngitis, rhinitis, pharyngitis, sinusitis, tonsilitis, laryngitis, tracheitis, laryngotracheitis |
| Bronchitis | Acute bronchitis |
| Influenza | Influenza |
| Pneumonia | Pneumonia |
| Asthma | Asthma |
| COPD | Chronic obstructive pulmonary disease, chronic bronchitis, emphysema |
| Pneumonitis | Pneumonitis |
| Fibrosis | Pulmonary fibrosis |
| Interstitial lung disease | Interstitial lung disease |

the reviewers and the wider study team. Studies which meet all the inclusion criteria and none of the exclusion criteria (see below) will be considered for full-text review. If the study cannot be unequivocally excluded during this phase, it will be considered for full-text review.

2. Full-text review: this step will be conducted independently by two reviewers in full. Any discrepancies will be resolved by discussion between the reviewers and the wider study team.

We will use the reference manager Endnote V.20 to store citations and screen all identified references.

### Selection criteria

The inclusion criteria for the systematic review will be:

1. The exposure is cancer survivorship. In this review, our focus will be on the survivorship period after initial active cancer treatment, so the exposed group must include follow-up time beyond the period of active treatment or the first year since diagnosis.
2. The focus is on survivors of adult cancer; thus we will only include studies where individuals were aged≥18 years at diagnosis.
3. The study evaluates at least one respiratory disease of interest as an outcome of interest (see table 1 for details).
4. The study involves a comparator cohort of cancer-free individuals or a general population estimate used to calculate indirect measures such as standardised mortality Ratios. No other criteria will be set for the comparator.
5. The study has an observational longitudinal cohort, matched cohort or case–control design.
6. The study provides a relative measure of effect for a respiratory disease of interest in individuals with cancer compared with cancer-free controls. Examples of relevant measures of effect are relative risk ratios, ORs,

rate ratios and hazard ratios. If studies provide sufficient information on measures of frequency for any of the respiratory diseases of interest measured in individuals with cancer and cancer-free controls to allow the calculation measure of effect between the groups, the study will also be included.

We will exclude:

1. Duplicate studies, defined as repeated articles, using the remove duplicate function of the software prior to conducting the manual screening. Any articles not identified by the software, will be removed manually by the reviewers.
2. Studies where the exposed group is exclusively composed of patients with cancer under active treatment as well as cancer cohorts whose average follow-up time is≤1 year.
3. Studies in which the exposed group includes individuals without cancer (eg, families and caregivers of the patient).
4. Intervention studies, cross-sectional studies and studies that do not use original data (such as review articles, comments, letters, editorials and protocols). Any review study will be excluded but flagged and the citation lists will be evaluated in order to identify potentially eligible studies that were not captured in our search.
5. Conference abstracts will be excluded if no corresponding full-text journal article is found as we consider that the information in an abstract will not be sufficient to complete our data extraction nor conduct an accurate bias assessment.

While articles will not be excluded on the basis of language, practical limitations mean that we will only evaluate those studies captured by an English search strategy. Articles in languages not spoken by the study team will be translated using translation software. We will exclude full-text articles which are written in non-roman alphabets due to the limitations of our translation software.

Reasons for exclusion will be collected in a cascade manner, meaning that papers that fail to meet any of the selection criteria will be automatically excluded and not further assessed. Where multiple papers are found to report data conducted on the same study population sample, these will be flagged, but not excluded. The results of these studies cannot be considered independent and thus the inclusion and/or interpretation of these studies will be considered on a case-by-case basis after discussion within the study team. Any inclusion or exclusion of studies of this type will be detailed in the data extraction phase and reported in the Results section of the report.

### Data extraction

Data will initially be extracted from included studies into a spreadsheet. Data extraction will commence in July 2022. At minimum, we will detail information on the study article, population sample description, study design, exposure, comparator and outcome definitions, follow-up, measures of effect, measures of frequency and

## Box 2    Data items for extraction

Data items:
Administrative:
⇒ Authors
⇒ Journal
⇒ Year of publication
⇒ Financial/funding sources
⇒ Conflict of interest reported
Data/design/population:
⇒ Country/countries of origin
⇒ Data source
⇒ Time period covered
⇒ Study setting
⇒ Study design
⇒ Sample size
Comparison group:
⇒ Selection criteria of the comparison group
⇒ Mean age of the comparison group
Exposure:
⇒ Selection criteria for exposed group
⇒ Definition of cancer survivorship
⇒ Time since cancer diagnosis (summary stats)
⇒ Cancer stage (summary stats)
Outcome:
⇒ Respiratory disease category
⇒ Respiratory disease definition
⇒ Duration of follow-up (if applicable)
Statistical methods:
⇒ Statistical method for the calculation of estimates
⇒ Statistical method for the calculation of the method of precision
⇒ List of confounders included in the adjusted analysis (if applicable)
Results
⇒ Measure of frequency + precision estimate (comparison group)
⇒ Measure of frequency + precision estimate (exposed group)
⇒ Measure of effect + precision estimate (crude)
⇒ Measure of effect + precision estimate (adjusted)

statistical methodology. The exact data items extracted are listed in box 2.

When multiple *different* respiratory disease conditions or cancer types are reported in the same study, we will record or calculate information for all cancer types and respiratory outcomes. When multiple estimates are reported for the *same condition* (eg, if probable and confirmed influenza were reported as separate outcomes), we will only record or calculate the measure of effect for the outcome corresponding to the stronger certainty of diagnosis (in our illustrative example, we would select confirmed influenza). For *each* outcome we will record both the crude and adjusted effect measures as well as the list of confounders included in the adjusted analysis. If the results are stratified by severity, we will record all severity categories for said outcome.

### Risk of bias assessment

We will use the Scottish Intercollegiate Guidelines Network (SIGN) methodology checklists as guidance to assess the risk of bias for each study design. The checklists are provided in online supplemental appendix 1. In brief,

we will assess if the study has a focussed aim, and whether there were opportunities for bias in the definition of the exposures and outcomes, selection of the study population and comparator group, and potential differential losses during follow-up. We will also assess how each study addresses confounding.

### Data analysis

We will initially present the key characteristics and results of all studies in descriptive tables. We will stratify the presentation of results by likely sources of heterogeneity such as cancer type and respiratory disease category (see table 1). If sufficient studies are found, we will also stratify by different population definitions (ie, if we find multiple definitions for cancer survivorship). We will meta-analyse groups of studies if we find sufficient studies that are sufficiently homogeneous in terms of population and outcomes for which the calculation of a summary measure would be meaningful. In the case that a summary estimate is calculated, we will use random-effects model. We will explore the consistency between studies by using the $I^2$ statistic.

### Patient and public involvement

No patients were involved in the development of this protocol

## ETHICS AND DISSEMINATION

To our knowledge, no comparable review has previously been conducted on respiratory outcomes in survivors of adult cancer. We aim to identify all available literature on a wide range of respiratory outcomes and cancer types considered, with no language, geographical or date limitations. This will provide a comprehensive picture of the spectrum of respiratory conditions that affect cancer survivors as well as understand if cancer-specific or respiratory condition-specific associations have been noted in the literature. Considering the high prevalence of respiratory disease in the general population, even a small increase in respiratory disease risk in cancer survivors will translate into an important impact on morbidity and mortality. The results of this review can be used to better understand respiratory morbidity in cancer survivors, identify evidence gaps and priorities for future research, and help devise possible mitigation strategies. Results will be disseminated to clinical audiences and may support clinicians and decision-makers in developing survivorship care guidelines.

While this protocol has been developed according to the PRISMA-P guidelines and has been registered on PROSPERO (ref: PROSPERO CRD42022311557), some potential limitations are worth noting. First, there is currently no universally accepted definition of cancer survivorship,[8] and this review focusses on the late respiratory challenges of cancer and its treatment. Consequentially, we have adopted the European Organisation for Research and Treatment of Cancer definition which

defines cancer survivor as 'individuals diagnosed with cancer who have completed their primary treatment (maintenance treatment can be ongoing)'.[9] We have used an inclusive approach towards this definition and will consider that the period of active treatment is expected to have been completed after 1 year since diagnosis, thus we will not exclude studies that include patients from cancer diagnosis with a mean follow-up of <1 year.[10]

Studies comparing long-term outcomes in cancer survivors and controls may be subject to a number of biases including selection bias, detection bias and competing-risks. We will evaluate the risk of bias in each included study in order to aid the interpretation of results, though we acknowledge that not all types of bias are specifically addressed by the SIGN checklists. Our review will also have other limitations. We may miss some relevant studies due to an imperfect search strategy, though hand searching of reference lists should identify additional papers and allow us to refine our search terms if needed. Finally, we are not including conference abstracts or preprints which could potentially exclude some relevant results.

Considering the high prevalence of respiratory disease in the general population, even a small increase in respiratory disease risk in cancer survivors will translate into an important impact on morbidity and mortality. The results of this review can be used to better understand respiratory morbidity in cancer survivors, identify evidence gaps and priorities for future research, and help devise possible mitigation strategies.

**Acknowledgements** We thank Ruchika Jain for agreeing to assist in conducting the screening of full-text articles.

**Contributors** KB, HC and KA designed the study. KA wrote the first draft of the protocol. JKQ provided clinical input for the selection of the study outcomes. All authors made substantial contributions to the manuscript, revised it critically for important intellectual content and read and approved the final version. KA is the guarantor of the review.

**Funding** KA is supported by the Medical Research Council Intercollegiate Doctoral Training Partnership Studentship (Grant no MR/N013638/1). KB is funded by a Wellcome Senior Research Fellowship (grant number 220283/z/20/z).

**Disclaimer** The funders had no role in developing this protocol.

**Competing interests** None declared.

**Patient and public involvement** Patients and/or the public were not involved in the design, or conduct, or reporting or dissemination plans of this research.

**Patient consent for publication** Not applicable.

**Provenance and peer review** Not commissioned; externally peer reviewed.

**ORCID iDs**
Kirsty Andresen http://orcid.org/0000-0002-5483-6482
Helena Carreira http://orcid.org/0000-0003-1538-2526
Jennifer K Quint http://orcid.org/0000-0003-0149-4869

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
