## [Reviewer comments · BMJ Open]

ARTICLE DETAILS

TITLE (PROVISIONAL)	Respiratory diseases in adult cancer survivors compared with the general population: a systematic review protocol.
AUTHORS	Andresen, Kirsty; Carreira, Helena; Quint, Jennifer; Bhaskaran, Krishnan

VERSION 1 – REVIEW

REVIEWER	Venkatesulu, Bhanu Loyola University Health System, Oncology
REVIEW RETURNED	29-Jul-2022

GENERAL COMMENTS	The authors have presented a protocol of the systematic review titled respiratory diseases in cancer survivors compared with the general population: a systematic review protocol. The protocol is an interesting study assessing the prevalence of respiratory disease in cancer patients as advances in cancer care improve cancer survivorship. The objectives are not focused, it will more reasonable if the authors report in terms of endpoints like incidence/prevalence of respiratory diseases in cancer patients; Either worsening of preexisting respiratory diseases or new onset respiratory diseases in cancer patients; hazard of death in patients with and without respiratory diseases in cancers. (This will be feasible obviously if the individual studies have data reported on this) 1. In the search strategy it is commonplace to include Cochrane database as well given Cochrane database has data from low-middle income countries2. In the selection criteria, how are the authors going to account for potential confounders like preexisting conditions like smoking, COPD, asbestos exposure which are causative and have association with both respiratory diseases and cancer
---

REVIEWER	Minotti, Chiara University of Padua
REVIEW RETURNED	08-Aug-2022

GENERAL COMMENTS	This systematic review protocol is overall very well-constructed, clear and complete. It was conducted according PRISMA guidelines, addressing all applicable items of the checklist and it is well-written. The scientific interest of the proposed topic is high. I recommend manuscript acceptance after minor revision, as follows: - the population should be better defined in the title, abstract, selection, inclusion criteria, methods, etc... and search strategy if applicable. The word "adult" is only mentioned once in the ethics and dissemination section. It is unclear whether papers on cancer
--

	survivors of all ages are to be included or not. If pediatric patients are excluded, please state that, specifying that only studies conducted on patients of age ≥ 18 years old will be included, and correct accordingly in the text. For example, Title: "Respiratory diseases in ADULT cancer survivors compared with the general population: a systematic review protocol", etc. If studies on patients of all ages are to be considered, please state that clearly. - please check introduction, screening and selection criteria sections for commonly used definitions or phrases and rephrase accordingly, to avoid plagiarism issues.
--	---

VERSION 1 – AUTHOR RESPONSE

Reviewer: 1

Dr. Bhanu Venkatesulu, Henry Ford Hospital

Comments to the Author:

The authors have presented a protocol of the systematic review titled respiratory diseases in cancer survivors compared with the general population: a systematic review protocol. The protocol is an interesting study assessing the prevalence of respiratory disease in cancer patients as advances in cancer care improve cancer survivorship.

The objectives are not focused, it will more reasonable if the authors report in terms of endpoints like incidence/prevalence of respiratory diseases in cancer patients; Either worsening of preexisting respiratory diseases or new onset respiratory diseases in cancer patients; hazard of death in patients with and without respiratory diseases in cancers. (This will be feasible obviously if the individual studies have data reported on this).

Thank you, objective 1 has been clarified to detail that outcomes of interest included incident diagnosis, hospitalisations, and mortality for each respiratory disease of interest:

1. **“Summarise the current evidence on the association between cancer survivorship and incident respiratory disease outcomes. We will consider the relative risk of incident diagnosis, exacerbations (considered as GP visits or hospitalisations) and mortality outcomes separately for each respiratory disease of interest between cancer survivors and the cancer-free population.”**

1. In the search strategy it is commonplace to include Cochrane database as well given Cochrane database has data from low-middle income countries

Thank you, we have added Cochrane to the search strategy.

2. In the selection criteria, how are the authors going to account for potential confounders like preexisting conditions like smoking, COPD, asbestos exposure which are causative and have association with both respiratory diseases and cancer

We plan to include all studies independent of potential confounding adjustments. However, we will extract information for both the crude estimate and adjusted estimate to understand the degree of confounding present in this population. Furthermore, we will include the adjustment factors used in the analysis as a data element in our data extraction form.

The following sentence has been added to the data extraction form:

“For each outcome, we will record both the crude and adjusted effect measures as well as the list of confounders included in the adjusted analysis”

Reviewer: 2

Dr. Chiara Minotti, University of Padua

Comments to the Author:

This systematic review protocol is overall very well-constructed, clear and complete. It was conducted according PRISMA guidelines, addressing all applicable items of the checklist and it is well-written. The scientific interest of the proposed topic is high. I recommend manuscript acceptance after minor revision, as follows:

- the population should be better defined in the title, abstract, selection, inclusion criteria, methods, etc... and search strategy if applicable. The word "adult" is only mentioned once in the ethics and dissemination section. It is unclear whether papers on cancer survivors of all ages are to be included or not. If pediatric patients are excluded, please state that, specifying that only studies conducted on patients of age ≥ 18 years old will be included, and correct accordingly in the text. For example, Title: "Respiratory diseases in ADULT cancer survivors compared with the general population: a systematic review protocol", etc. If studies on patients of all ages are to be considered, please state that clearly.

Added clarifications to specify that the review will be conducted in survivors of adult cancers in the title, abstract and selection criteria

- **“adult” added to title**
- **We have changed the term “cancer survivors” to “survivors of adult cancer” or “adult cancer survivors” in the abstract**
- **The following criterion has been added to the screening criteria “The focus is on adult cancer survivors; thus we will only include studies where individuals were aged ≥ 18 years at diagnosis”**

Note: We have clarified within the selection criteria that our definitions of a comparator group of cancer-free individuals will also include general population estimates used to calculate indirect measures such as Standardised Mortality Ratio.

- please check introduction, screening and selection criteria sections for commonly used definitions or phrases and rephrase accordingly, to avoid plagiarism issues.

Amended the following within the text.

Furthermore, cancer can weaken the immune system which may increase the risk of respiratory infections ~~It is thought that the effects of cancer and its treatment may increase the risk of respiratory infections due to weakened immune systems (5).~~ Additionally, cancer survivors often have a higher health burden. A quarter of cancer patients have pre-existing respiratory comorbidities ~~Furthermore, a quarter of cancer patients have been diagnosed with a pre-existing respiratory comorbidities (6)~~

Reviewer: 1

Competing interests of Reviewer: None

Reviewer: 2

Competing interests of Reviewer: I declare no competing interests.

VERSION 2 – REVIEW

REVIEWER	Venkatesulu, Bhanu Loyola University Health System, Oncology
REVIEW RETURNED	23-Sep-2022
GENERAL COMMENTS	My comments are addressed and the manuscript is acceptable for publication